# Enhanced Scratch Performance of Plasma Sprayed Hydroxyapatite Composite Coatings Reinforced with BN Nanoplatelets

**Yao Chen *** , **Jia Ren, Weiwei Liu and Dong Zhao**

School of Mechanical & Electric Engineering, Soochow University, Suzhou 215021, China;
rj13862134680@163.com (J.R.); liuweiwei@suda.edu.cn (W.L.); zhaodong@suda.edu.cn (D.Z.)
* Correspondence: chenyao@suda.edu.cn

**Abstract:** In recent years, research on hydroxyapatite (HA) coatings has been driven by the demands of clinical applications. However, the intrinsic brittleness of HA limits its potential in the use for the load-bearing implant. To improve mechanical properties of the HA coating itself, a HA composite coating reinforced with hexagonal boron nitride nanoplatelets (BNNP) was fabricated using plasma spray, and its scratch behavior was investigated in this research. Typical brittle fractures such as microcracks both in and beyond the residual groove and material chipping were observed in the HA coating, while stronger and tougher BNNP/HA coatings exhibited a dominant role in protecting them from scratch damage through resisting plastic deformation and brittle microfracturing. Moreover, easier grain sliding within a splat and splat sliding at the splat boundaries due to the presence of BNNPs, and the nature porosity at different length scales of the as-sprayed HA composite coatings would provide significant self-lubricating effects to reduce the lateral force during scratching and alleviate the contact damage. Therefore, the addition of BNNPs renders HA coating with low scratch friction and enhanced tolerance to surface damage, which is naturally beneficial for the long-term durability and reliability of the implants.

**Keywords:** scratch resistance; BN nanoplatelet; hydroxyapatite; coating; plasma spray

## 1. Introduction

Titanium alloys with a combination of high toughness and specific strength have been widely used for orthopedic implants [1]. However, the low wear resistance and the nature of the bioinert-induced inferior adhesion bonding with the bone tissues are well known to impair their long-term clinical performance. In addition, these metallic materials usually release harmful metallic ions and lead to adverse effects on the surrounding tissues when they get in contact with body fluids [2]. Therefore, the coating of hydroxyapatite (HA) deposited on the surface of a metallic implant has been increasingly recognized as an attractive and effective approach to promote its osteointegration and interfacial bonding with the surrounding bone cells [3,4] because HA is a major inorganic constituent of natural bone and exhibits excellent bioactivity/osteointegration properties. Unfortunately, poor fracture toughness and wear resistance of monolithic HA limit its potential applications as a coating material, especially on load-bearing metallic implants [5,6]. To achieve improved mechanical properties of HA coating, HA composite coatings reinforced with a second phase, such as alumina ($Al_2O_3$) [7,8], yttria-stabilized zirconia (YSZ) [9,10], and titania [11,12] have been considerably developed in the past two decades. Despite significant improvement achieved in fracture toughness of these HA composite coatings, it should be noted that the addition contents of these ceramic particles, basically biologically inert, are usually as high as 30–50% in weight fraction (wt.%), inevitably leading to degradation of the excellent bioactivity and osteointegration associated with HA.

In recent years, carbonaceous nanomaterials such as carbon nanotubes (CNTs) and graphene have been considered as promising nanofillers in ceramics to improve their fracture toughness [13,14] due to their exceptional mechanical properties. Meanwhile, graphene nanosheets (GNSs) have been demonstrated to render HA composites with ~80% improvement in fracture toughness even at 1.0 wt.% GNS concentrations [15]. Nevertheless, it is noteworthy to note that lower oxidation temperature (~400–450 °C) of carbonaceous nanomaterials [16] does not make them suitable as nanofillers in composites/coatings for high-temperature applications or synthesized through the high-temperature processes. The very recent burst into the research scene of hexagonal boron nitride nanoplatelets (BNNPs), possessing similar two-dimensional structures and comparable mechanical properties (e.g., tensile strength ~35 GPa, elastic modulus 700–900 GPa) [17] to those of graphene, have created a great expectation about their promising benefits on their enhanced fracture toughness of ceramics. Most importantly, BNNPs can withstand elevated temperatures of up to ~950 °C [18], making them more attractive for being toughening nanofillers in HA coatings fabricated through plasma spray—a US Food and Drug Administration (FDA) approved technique for depositing HA coatings on implants for clinical applications [19]. Furthermore, BN nanotubes (BNNTs) in a HA composite were reported to improve its toughness without compromising the proliferation and viability of the osteoblasts [19], implying that BNNPs are biocompatible in nature. Additionally, compared to the black color along with graphene, the white color of BNNPs matches well with HA from the viewpoint of visual impression. Therefore, BNNP-reinforced HA composite coatings have been successfully fabricated using plasma spray in our previous research [20].

It is well known that the coated implants stay inside the human body for a long time. Thus, wear debris generated from the coated surface during the service period can either be introduced into the human body causing a toxic effect or be trapped in the same zone and play the role of an abrasive third body, leading to severe wear damage of the implant [2,21]. Moreover, most wear damage of a solid material usually initiates at small length scales [21]. Hence, an in-depth understanding of microscale wear behavior of BNNP/HA composite coatings is critical to explore its potential clinical applications.

The aim of the present work, as an extension of our previous research, was to investigate the influences of the addition of BNNPs on the surface damage behavior and the microscale tribological properties of HA-based composite coatings.

## 2. Materials and Methods

HA nanorods with a diameter of ~20 nm and a length of ~100 nm (Nanjing Emperor Nano Material, Nanjing, China) and BNNPs with a thickness of ~20–30 nm and a diameter of ~0.5–5 μm (Nanjing XianFeng Nano Material Company, Nanjing, China) were employed as precursor materials. To ensure homogeneous distribution of BNNPs in the HA composite coatings, the as-received BNNPs were ultrasonicated for 3 h in ethanol with a concentration of about 0.1 mg/mL. Then, HA nanorods were added into BNNP suspension followed by 1 h ultrasonication and 3 h magnetic stirring, respectively. Finally, the obtained composite powders were dried in an oven at 80 °C for 24 h. The compositions chosen here were pure HA, 1.0 wt.% BNNP/HA (HA-1B), and 2.0 wt.% BNNP/HA (HA-2B). To obtain microsized agglomerates with good flowability for plasma spray, spray drying was employed to obtain porous spherical nanostructured agglomerates with a diameter of 40–75 μm using a spray drier (LGZ-8, Wuxi Dongsheng Spray-Granulating, and Drying Equipment Plant, Wuxi, China). Prior to plasma spray, Ti-6Al-4V substrates (100 mm × 15 mm × 5 mm) were blasted using $Al_2O_3$ particles with an average size of ~1 mm and followed by being cleaned with acetone. These spherical spray-dried agglomerates were plasma sprayed using SG 100 gun (Praxair Surface Technology, Danbury, CT, USA) on Ti-6Al-4V substrates, in which 20–25 kW plasma power and 600–700 A plasma-gun current were used with a powder feeding rate of 4.5 g/min and a standoff distance of 100 mm. Argon was used as the primary gas (flow rate: 32 slpm) with helium as an auxiliary gas (28 slpm). Argon was also used as a powder carrier gas (8 slpm).

Prior to scratch tests, the as-sprayed samples were ground by 800-grit silicon carbide abrasive paper and followed by polishing with diamond particles size of 1 μm. Instrumented scratch tests were conducted using a micro-scratch Tester (CSM Instruments, Peseux, Switzerland) equipped with tangential friction force sensor and penetration depth sensors. A spherical diamond tip with a radius of 100 μm was utilized under a constant load of 2 and 5 N, a scratching speed of 800 μm/min, and a scratch distance of 800 μm were employed. At least three scratches, 500 μm apart, were made on each sample to ensure experimental reproducibility. The residual scratch grooves were characterized using scanning electron microscopy (SEM, Hitachi S-4700, Tokyo, Japan). Micro-Raman spectroscopy (Renishaw, London, UK) was employed to confirm the existence of BNNPs on the scratch grooves of the composite coatings with an Argon ion laser of wavelength 532 nm and an acquisition time of 10 s.

## 3. Results and Discussion

The deposited coatings were observed to have a uniform thickness of ~200 μm and free of microcracks; all as-sprayed coatings consisted of HA, TCP, CaO, and TTCP [20]. In general, HA usually decomposes into a mixture of TCP and TTCP in the temperature range of ~1000–1450 °C [22]. The presence of the CaO phase in all as-sprayed coatings implied that TTCP partial decomposition occurred, i.e., $Ca_4(PO_4)_2O \xrightarrow{1550-1630\ °C} 4CaO + P_2O_5$, which is likely due to high-temperature exposure of feedstock materials during the plasma spray. Meanwhile, $P_2O_5$ should be eliminated from the coatings because of its low boiling point (~380 °C).

SEM images of the polished cross-section of the as-sprayed HA coatings are depicted in Figure 1. It is clear that the coating-substrate interfaces were clear, and no microcrack could be observed near the interface, implying that the adhesive strength of the coating to the substrates is high. Owing to the porous nanostructured agglomerates, the heat transfer toward the center of the agglomerates during plasma spray might be significantly limited, leading to the presence of partially melted/unmelted zones. Subsequently, these partially melted/unmelted zones present in the HA coatings, followed by the grinding and polishing in the sample preparation, resulted in the observed craters, as pointed out by white circles in Figure 1.

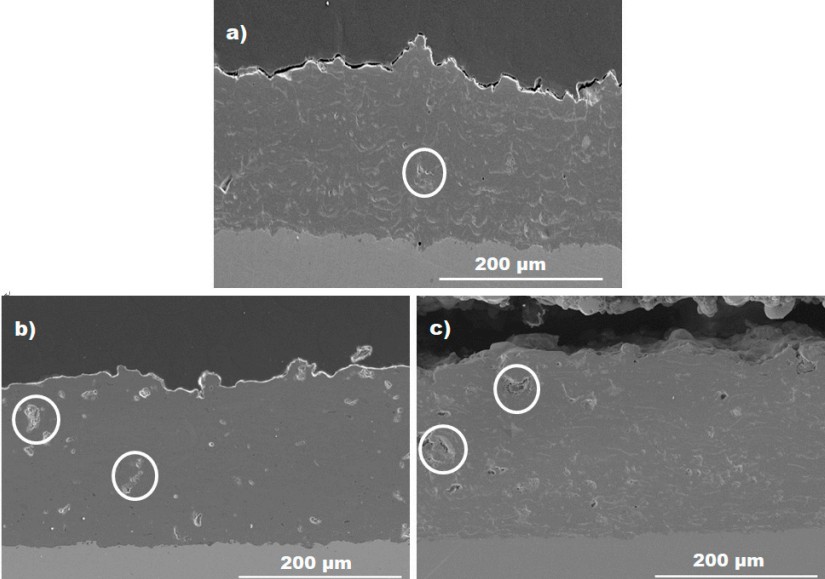

**Figure 1.** SEM images of cross-sectional view of (**a**) HA coating, (**b**) HA-1B coating, and (**c**) HA-2B coating.

As clearly shown in Figure 2, the typical surface image of the as-sprayed coatings, the surfaces of these coatings consisted of partially melted and un-melted particles that were all adhered to

the well-melted splats, in which small spherulites and dune-shaped particles corresponded to the un-melted and partially melted particles, whereas smooth surfaces corresponded to fully melted splats.

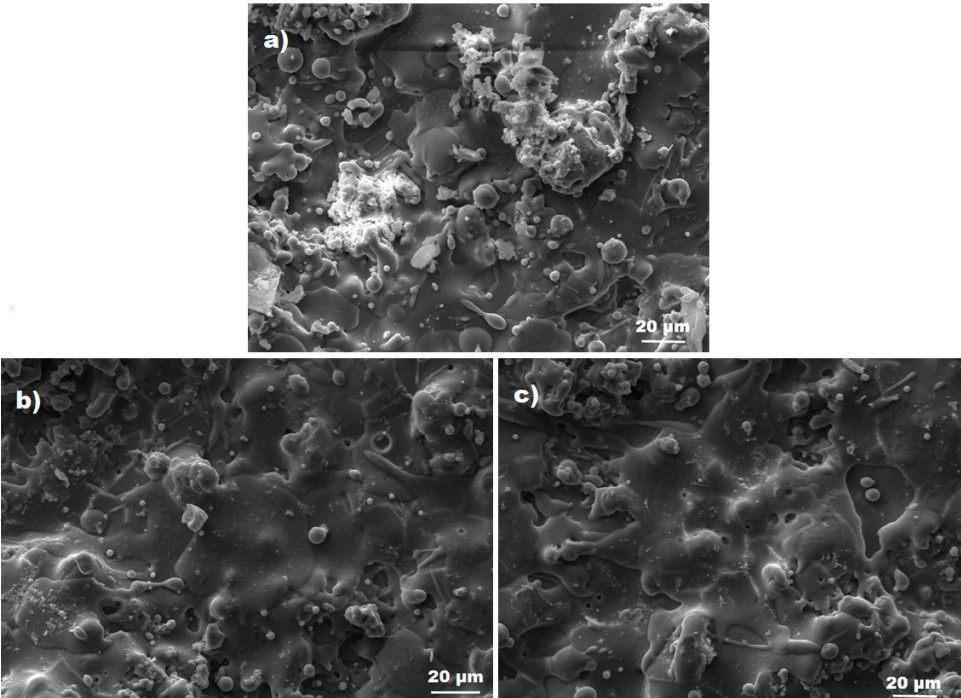

**Figure 2.** SEM images showing the surface morphologies of (**a**) HA coating, (**b**) HA-1B coating, and (**c**) HA-2B coating.

When scratching the brittle materials, the primary damage mechanisms should be as follows: plastic deformation, followed by cracking, and finally chipping [23]. SEM micrographs of the residual scratch tracks of the as-sprayed coatings at a normal load of 2 N are shown in Figure 3—it is clear that the lower sliding load produced relatively smoother track surfaces. In the case of HA coating, HA grains within the scratch grooves readily plastically deformed, some microcracks were visible in and beyond the scratch groove, and localized chipping occurred around the scratch boundaries (Figure 3a), implying that plastic deformation and slight brittle fracture dominated the scratch behavior of HA coating at this load. In contrast, the entire scratched surface of the HA-1B coating was basically smooth without noticeable microcracks (Figure 3b), and HA-2B coating was also relatively smooth with little grains being pulled out (Figure 3c). These characteristics imply that the predominant scratch mechanism of the BNNP/HA coatings was elastic-plastic deformation.

An increase in the applied load (5 N) allowed some distinguishing surface damage phenomena, as shown in Figure 4a. Among them, the as-sprayed HA coating displayed severe damage such as microcracks both in and beyond the residual groove, microfracturing/chipping at the groove boundary (Figure 4a). In the case of as-sprayed BNNP/HA coatings, it is evident from Figure 4b,c that the scratch width decreased, and the material chipping at the groove edges was inhabited significantly.

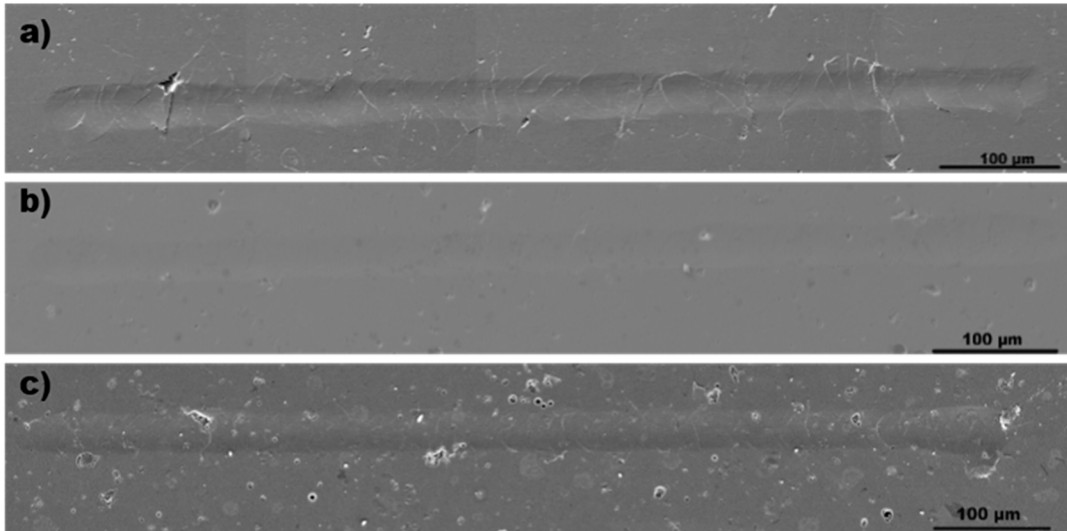

**Figure 3.** SEM images showing residual groove morphologies of (**a**) HA coating, (**b**) HA-1B coating, and (**c**) HA-2B coating at a normal load of 2.0 N.

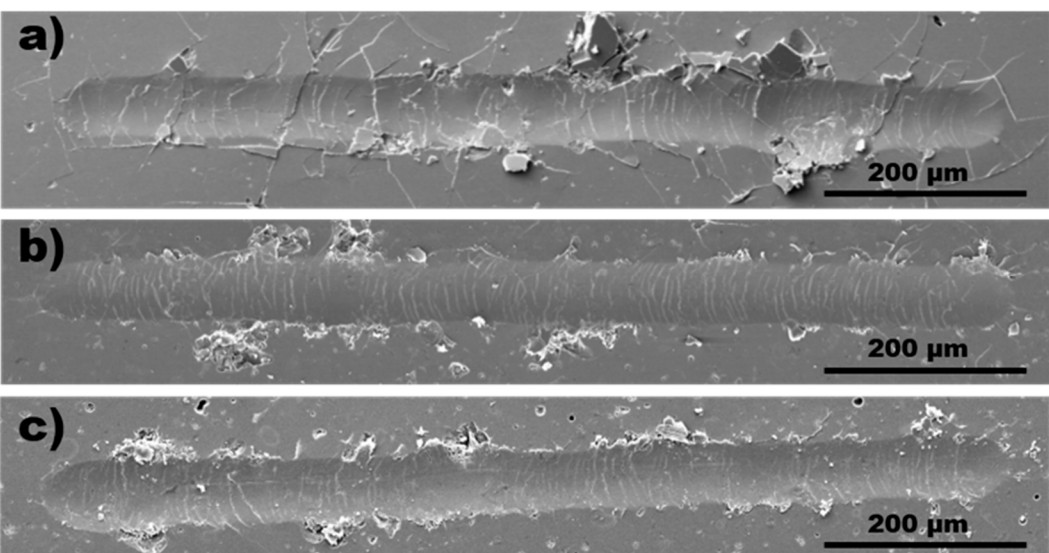

**Figure 4.** SEM images showing residual groove morphologies of (**a**) HA coating, (**b**) HA-1B coating, and (**c**) HA-2B coating at a normal load of 5.0 N.

The residual scratch depth is usually considered as a measure of surface damage after elastic recovery. As depicted in Figure 5a, it is clear that residual scratch depths for the as-sprayed HA coatings at a load of 2 N were almost the same (in the range of 0.5–0.9 μm) regardless of the addition of BNNPs. However, at a load of 5 N, the residual scratch depth of the HA coating increased significantly to 3.0–4.5 μm, while the residual scratch depths for the HA-1B coating and HA-2B coatings were on the order of 1.5–2.5 and 2.0–3.5 μm, respectively (Figure 5b). On the basis of the well-accepted fact that the splat thickness of plasma sprayed coatings is ~2–3 μm [24], it is reasonable to conclude that plastic deformation of the as-sprayed HA coatings during scratch at a load of 2 N occurred within the single splat, while the coatings deformed plastically in the region involving several splats at a load of 5 N. Plasma-sprayed coatings usually display a lamellar structure due to the nature of the splat stacking and the adjacent splats bonding together at different length scales intercepted by voids, inter-splat pores, and microcracks [25], allowing poor adhesion between the adjacent splats and easier splat sliding [26]. Our previous research reported that some BNNPs embedded at the splat boundaries would maximize their surface area getting in contact with the adjacent splats to consolidate splat

boundaries in virtue of their outstanding mechanical properties and two-dimensional structure [20]. Therefore, it is deduced that the BNNP/HA coatings are capable of resisting plastic deformation when interacting with a sliding rigid ball under higher applied load.

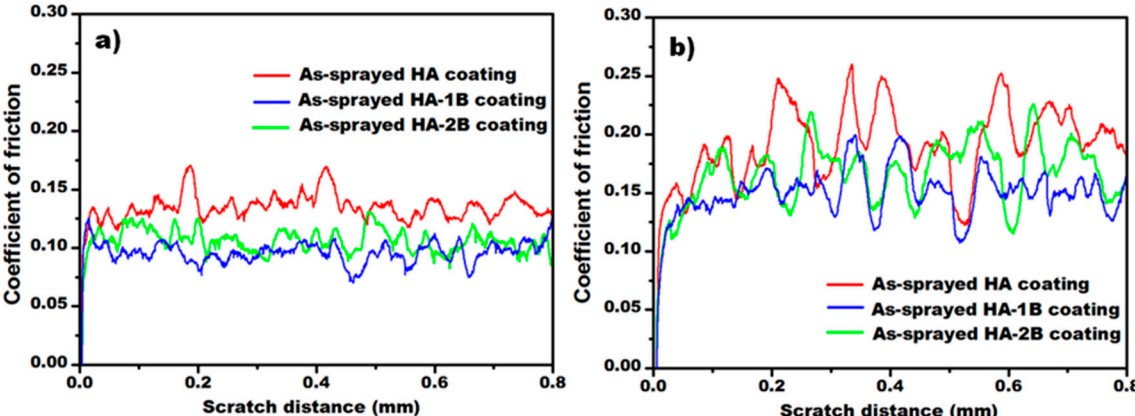

**Figure 5.** Residual scratch depths of the as-sprayed coatings during scratch at a load of (**a**) 2 N and (**b**) 5 N.

Variations in coefficients of friction (CoF) for the as-sprayed coatings at different loads are shown in Figure 6. It is clear that, as compared to HA coatings, the BNNP/HA coatings exhibited a substantial reduction in the friction coefficient by dropping ~20% at a load of 2 N and ~25% at a load of 5 N. This strongly manifests significant self-lubricant effects associated with the added BNNPs took operative during the scratching. Nevertheless, previous research [21] on the scratch behavior of graphene nanoplatelets (GNP), reinforced through alumina composites fabricated through spark plasma sintering (SPS), found that their CoFs were independent of the addition of GNPs because these GNPs were constrained by the surrounding grains in a dense structure so as not to activate grain sliding. In the case of plasma-sprayed coatings, incomplete melting and improper stacking of splats usually led to such defects, such as voids, pores, and microcracks at the splat boundaries, resulting in the porous nature at different length scales. Furthermore, these BNNPs, soft in the out-of-plane direction, is expected to facilitate the grain sliding in a splat and splat sliding, which is responsible for the self-lubricant effect.

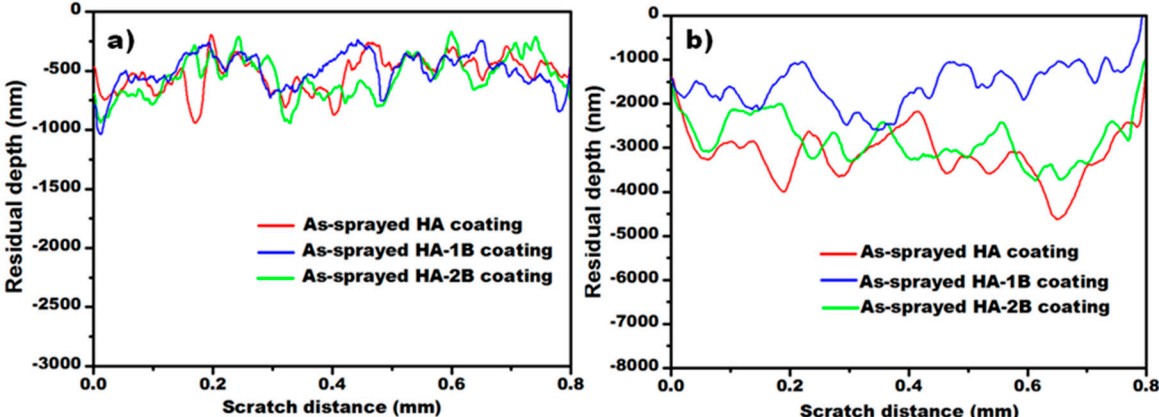

**Figure 6.** Coefficient of friction profiles of the as-sprayed coatings during scratch at a load of (**a**) 2 N and (**b**) 5 N.

It is evident from Figures 5 and 6 that the residual depths and CoFs of as-sprayed HA coatings in the process of scratching fluctuated strongly, especially at higher loads of 5 N. These characteristics should likely be due to the presence of partially melted/unmelted zones on the coating surface,

which inevitably led to the formation of craters during the grinding and polishing in the sample preparation prior to the scratching tests. As such, once the diamond tip comes in contact with these cavities during scratching, sudden changes in the penetration depth and CoF occur. Moreover, as shown in Figures 5 and 6, the residual scratch depth and CoF curves of all the sprayed coatings at a higher normal load of 5 N displayed a large degree of localized fluctuations relative to those at a lower load of 2 N. Generally, scratch testing is by its nature performed on the small volumes beneath the rigid tip, and the deformation volume increases with the applied load. Hence, the deformation region during scratching at higher loads would contain much more craters and microstructure defects along with plasma-sprayed coatings such as voids and inter-splat pores, leading to larger oscillations in scratch residual depth and CoF.

Surface damage of a tested solid material during scratching is significantly dependent on whether the deformation resulting from the applied contact stresses could be effectively mitigated or not. In order to estimate the generated stress levels relating to the damage caused, the Hertzian contact pressure (*HCP*) and the shear stress (τ) were calculated using the following equations [27].

$$HCP = \left( \frac{6F_n(E^*)^2}{3.14^3 \times R^2} \right)^{\frac{1}{3}} \tag{1}$$

$$T = HCP \times COF \tag{2}$$

in which

$$\frac{1}{E^*} = \frac{1 - v_1^2}{E_1} + \frac{1 - v_2^2}{E_2} \tag{3}$$

The calculated shear stress for the HA sample, HA-1B coating, and HA-2B coating sample at 2 N were ~1.33, ~1.20, and ~1.07 GPa, respectively. When the applied load increased (5 N), the shear stress increased to be ~2.64, ~2.40, and ~2.18 GPa for the HA coating, HA-1B, and HA-2B coatings, respectively. The reductions in shear stress, i.e., ~9.8%–19.5% at 2 N and ~9.0%–17.4% at a load of 5 N, might ascribe to the self-lubrication of composite coatings.

To understand the role of BN nanofillers in the scratch behavior, the residual scratch grooves at a load of 5 N for the as-sprayed samples were further investigated at higher magnification using SEM, and their images are presented in Figure 7. It is evident from Figure 6a that HA coating displayed typical brittle damage, in which microcrack propagation and microfracturing presented at the scratch boundaries and these microcracks advanced into the unscratched zones. Such surface damage of the HA coating would be detrimental to the coating integrity, making it a more rapid failure. Furthermore, some scratch debris with a size of 20–30 μm was observed to attach on the scratch grooves (Figure 7a), these accumulated debris would, in turn, serve as third-body abrasives to accelerate the coating damage. In contrast, the residual scratch surface morphologies of the as-sprayed BNNP/HA coatings, as shown in Figure 7b,c, depicted that the microcrack extension into the unscratched regions and brittle microfracturing were greatly inhibited, and the size of the scratch debris was also decreased to several microns, indicating that surface damage was less severe. As shown in Figure 7d, BN nanofillers were observed to present in the cavity at the scratch boundaries where microfracturing occurred. These BNNPs are capable of interlocking the fractured fragments to inhibit them from being involved on the scratch track as third-body abrasives, subsequently leading to the slow-down of surface damage. The enhanced tolerance to surface damage of the composite coatings should be ascribed to the unique synergetic strengthening and toughening effects induced by the added BNNPs. Our previous research reported the measured mechanical properties of HA and BNNP/HA coatings by the indentation technique [20], as listed in Table 1. As compared to the HA coating, the indentation hardness and elastic modulus of the HA-1B coating increased. Nevertheless, further BNNP, adding up to 2 wt.%, led to a slight decrease in both the hardness and elastic modulus of the HA composite coating. The fracture toughness of as-sprayed HA-1B coating showed a ~29.8% improvement (from 0.57 ± 0.12 to 0.74 ± 0.22 MPa m$^{1/2}$), and it increased up to 0.80 ± 0.25 MPa m$^{1/2}$ for HA-2B coating. Moreover,

the brittleness index (*BI*), defined as the $H/K_{IC}$ ratio of the tested material, is often used to represent its tolerance to contact damage during scratch and wear because *BI* reflects the combined response of the tested materials rather than taking either *H* or $K_{IC}$ into account separately [28]. The calculated *BI* for the HA coating was ~6.25, and it decreased to ~5.46 for the HA-1B coating and ~4.55 for HA-2B coating. As such, the inclusion of BNNP nanofillers makes HA composite coatings stronger and tougher, being more tolerant of brittle damage from scratching. The presence of BNNP appearing on the scratch groove was evidenced by Raman spectra, as shown in Figure 8, in which the characteristic G band frequency was ~1365 cm$^{-1}$, corresponding to the $E_{2g}$ phonon mode of BN. It should be noted that the penetration depth of the laser photon is limited at the near-surface of solid material (in carbon: 20–50 nm; in amorphous Si: 10–100 nm) [29]. However, the BNNPs were not observed, even when using high-magnification SEM. Hence, these detected BNNPs might locate to the near-surface of the scratch groove of the composite coatings due to the fact that these BNNPs with high flexural strength easily bend and tilt towards the surface or subsurface of the wear track when they get in contact with the counterpart under repeated rolling [30].

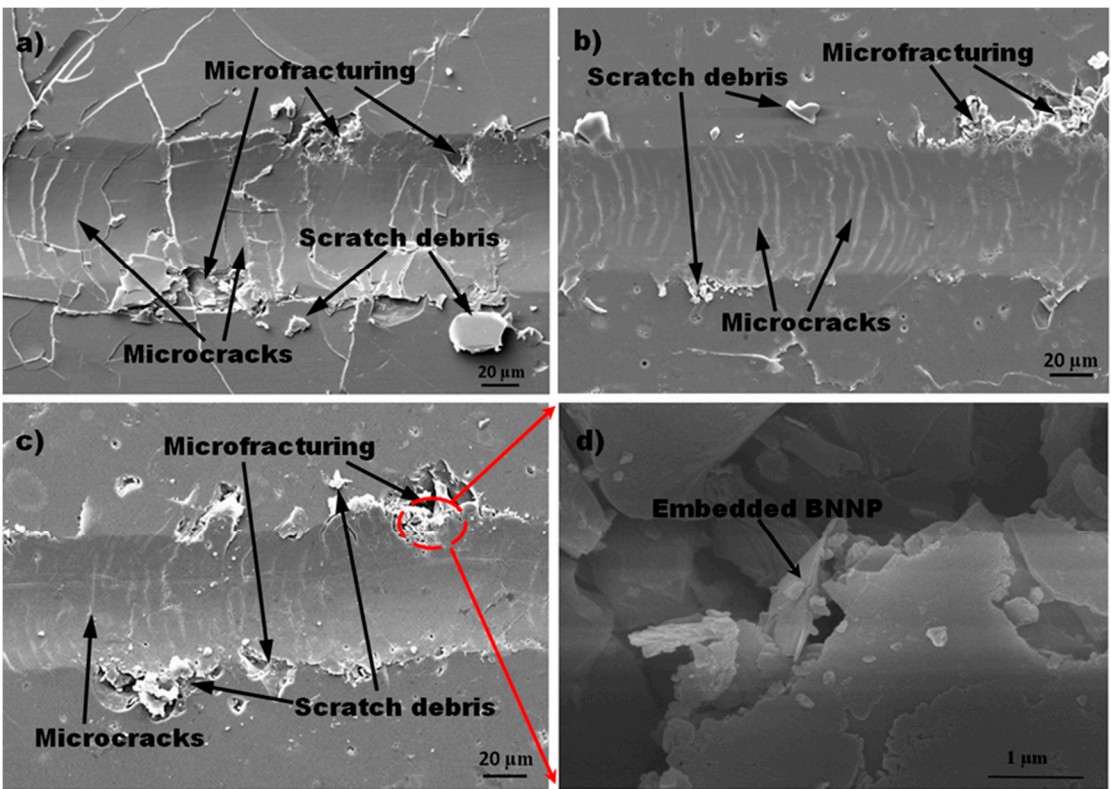

**Figure 7.** High-magnification SEM images showing residual groove morphologies of (**a**) HA coating, (**b**) HA-1B coating, (**c**) HA-2B coating at a load of 5 N, and (**d**) BNNPs embedded in cavity at scratch boundary.

**Table 1.** A summary of the mechanical properties of the as-sprayed HA coatings.

| Coatings | *E* (GPa) | *H* (GPa) | *KIC* (MPa m$^{1/2}$) | *Brittleness Index (BI)* |
|---|---|---|---|---|
| HA | 62.97 ± 3.95 | 3.45 ± 0.23 | 0.57 ± 0.12 | 6.25 ± 1.08 |
| HA-1B | 74.39 ± 3.70 | 3.76 ± 0.26 | 0.74 ± 0.22 | 5.46 ± 1.27 |
| HA-2B | 71.23 ± 2.75 | 3.72 ± 0.30 | 0.80 ± 0.25 | 4.55 ± 0.72 |

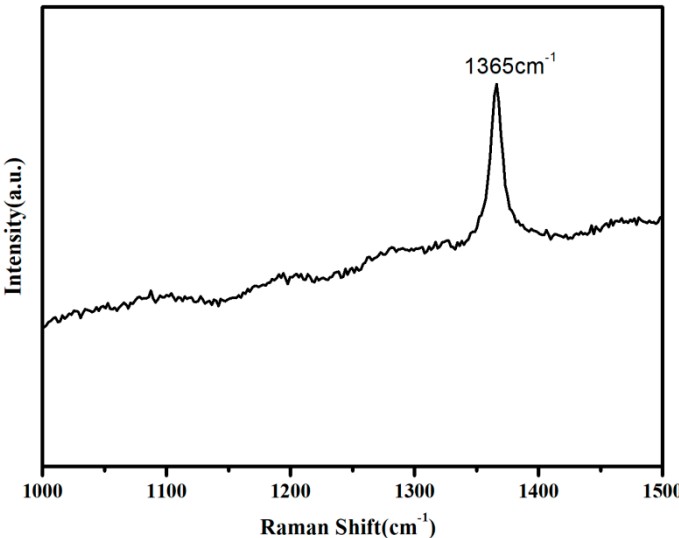

**Figure 8.** Raman spectra of the scratch groove of HA-1B at a load of 5 N.

However, it is necessary to note that local microfracturing, along with the HA-2B coating was more severe with respect to that of the HA-1B coating (Figure 7). This is likely due to BNNP agglomeration occurring in the HA-2B coating because the homogeneous distribution of BNNPs in the composites is still challenging when high addition content of BNNPs is employed. As shown in Figure 9a, a typical wrinkle feature on the surface of pulled-out BNNP was observed, which is expected to help in forming stronger interfacial bonding through mechanical locking between a BNNP and HA grains. Nevertheless, BNNP agglomeration occurred in the HA-2B coating (Figure 9b). BNNP agglomeration not only acts as stress concentration for crack nucleation and propagation but also induces porosity and inter-layer splitting to reduce the load transfer efficiency from the HA matrix to the inner BNNP layers, subsequently resulting in deteriorations in the toughness and strength of the HA-2B coating. Hence, BNNP agglomeration made the HA-2B coating more liable to surface damage.

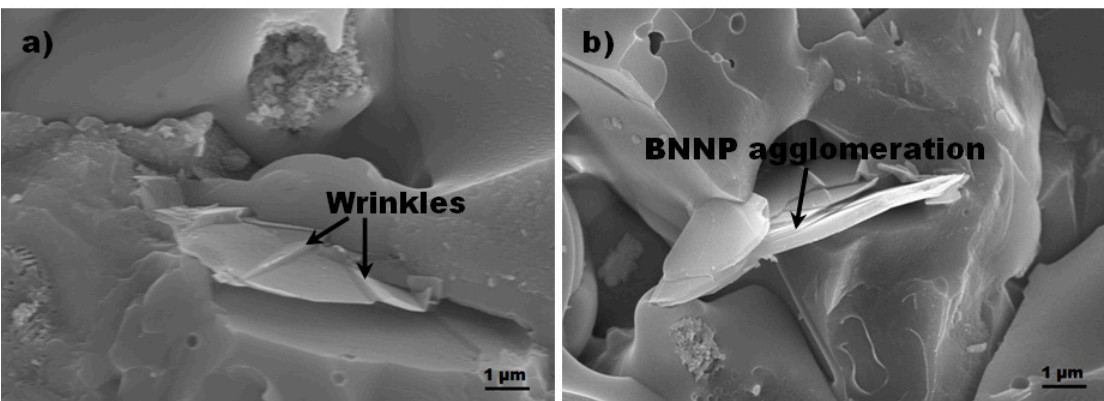

**Figure 9.** SEM images showing the BNNPs locating on the fracture surfaces of (**a**) HA-1B coating and (**b**) HA-2B coating.

It is evident from Figures 3, 4 and 7 that local microfracturing or material chipping occurred preferentially at the scratch boundaries regardless of the addition of BNNPs. It is well recognized that the localized hydrostatic compressive stress should be produced within the scratch groove when the tested material gets in contact with the rigid sliding tip, and this localized hydrostatic compressive stress is high to impede cracking and brittle fracture to some extent [31]. However, the compressive pressure on the groove edges would decrease to near zero so as not to impede cracking and fracture, leading to preferential microcracking or microfracturing around the scratch edges. More interestingly,

as compared to the HA coating, the increased density of microcracks within the residual grooves of BNNP/HA coatings was identified in Figures 4 and 7. The authors think the following factors should be responsible for the following characteristics: (i) during scratching, tensile frictional stress exists in the coating surface behind the trailing edge of the spherical tip, leading to tensile cracking. (ii) coefficient of thermal expansion (CTE) of BNNP ($-2.9 \times 10^{-6}$/K along the $\alpha$ axis at ambient temperature) [32] is reported to be much lower than that of HA ($11.5 \times 10^{-6}$/K) [33], and; therefore, BNNPs would expand while the HA matrix would shrink during the cooling process after plasma spray, subsequently leading to hoop compressive stresses applied upon these BNNPs and tensile stresses on the HA grains. Therefore, the above tensile stresses on the coating surface during scratching resulted in increased microcrack density within the scratch grooves of the BNNP/HA coatings, which dissipated elastic strain energy that would otherwise cause fracturing of the scratched sample.

## 4. Conclusions

In summary, the scratch performance of HA composite coatings containing BNNPs was investigated using spherical instrumented scratch tests. Benchmarked against typical cracking and microfracturing observed in the HA coating, BNNP/HA composite coatings exhibited enhanced resistance to microcrack extension and brittle microfracturing. Moreover, the added BN nanofillers rendered the composite coatings with self-lubricating effects to alleviate contact damage. The residual stress resulting from the mismatch of the thermal expansion between the BNNPs and HA matrix accelerated microcrack formation within the scratch grooves of the composite coatings, which contributed to the dissipation of elastic strain energy to impede microfracturing. The BNNP/HA composite coating with improved scratch resistance is expected to be naturally beneficial for the long-term durability and reliability of the implants.

**Author Contributions:** Conceptualization, methodology, writing—original draft preparation, writing—review and editing, project administration, funding acquisition Y.C.; methodology, data curation, visualization, writing—original draft preparation, J.R.; methodology, visualization, W.L.; methodology, data curation, D.Z. All authors have read and agreed to the published version of the manuscript.

**Funding:** This research was financially supported by the National Natural Science Foundation of China (51471113).

**Conflicts of Interest:** The authors declare no conflict of interest.

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
