# Peer review of "Enhanced Scratch Performance of Plasma Sprayed Hydroxyapatite Composite Coatings Reinforced with BN Nanoplatelets"

_coatings, doi:10.3390/coatings10070652_

Round 1

Reviewer 1 Report

The paper is topical, the field of biocompatible materials for implants is the goal of many researches, as evidenced by an overview of the references used.  

I have the following comments:  

1. Line 87-88, the details of the plasma spraying experiments were provided elsewhere [20]?   After all, it would be appropriate to present the parameters of plasma spraying, the distance of the samples from the plasma nozzle, the number of layers, as well as the test samples, or sample preparation strategies. The reader should have this information in the article. Even if the parameters were in accordance with the parameters in another paper, it is necessary to declare them in this paper for the completeness of the data. A scientific contribution is a product for which access is paid for. The reader does not have to search some data. parameters, or buy more articles to search missing data. The article is not so extensive that it is not possible to supply the complete data and parameters. I recommend to do so. It would also be appropriate to show the basic scheme of testing. I leave it to the authors' discretion.

Author Response

Reviewer #1:

The paper is topical, the field of biocompatible materials for implants is the goal of many researches, as evidenced by an overview of the references used.   

I have the following comments:  

1. Line 87-88, the details of the plasma spraying experiments were provided elsewhere [20]?   After all, it would be appropriate to present the parameters of plasma spraying, the distance of the samples from the plasma nozzle, the number of layers, as well as the test samples, or sample preparation strategies. The reader should have this information in the article. Even if the parameters were in accordance with the parameters in another paper, it is necessary to declare them in this paper for the completeness of the data. A scientific contribution is a product for which access is paid for. The reader does not have to search some data. parameters, or buy more articles to search missing data. The article is not so extensive that it is not possible to supply the complete data and parameters. I recommend to do so. It would also be appropriate to show the basic scheme of testing. I leave it to the authors' discretion.

Response: The authors agree with the reviewer’s comment. The plasma spray processing parameters has been provided in the revised manuscript.

Reviewer 2 Report

At the moment, the topic is of interest and the article is written in a professional manner. However, the reviewer has identified some important issues that are not found in the article at this time.

1. Considering the layers thickness of ~ 200 μm, in order to highlight correctly the aspects indicated in the text, the results of some analyzes performed in samples cross-section must be presented. These results can highlight the fracture mode and the depth to which they produce the effects of the force action. It is important whether the fracture is maintained in the ceramic layer or occurs in depth and even induces detachment from the metal substrate.

2. There are many possible logical questions for any potential reader of the article. What were the considerations used to choose the 2N and 5N forces and why higher values were ​​not used? At what value does the exfoliation of the ceramic layer occur? What are the values ​​of similar forces when inserting implants in cortical bone and trabecular bone?

3. Prior to the scratch tests, the samples were ground by 800-grit silicon carbide abrasive paper and followed by polishing with diamond particles to allow the test to be made. In real conditions of use (referred to in the introductory part of the article), this is not possible, as the implant surfaces have very complex geometries. Moreover, generally, at macro level the aim is to obtain surfaces with the highest possible roughness. Given the above, how relevant is the performed test?

4. In the reviewer's opinion, performing strictly the scratch tests is not enough to achieve the proposed objectives. Performing ball bearing wear tests on the disc would have been eloquent.

5. For a correct comparison of the results, in Figure 1 the analysis corresponding to the HA-1B sample, must be introduced.

6. It is not clear how the measurement of the variations in coefficients of friction (CoF) was performed for the as-sprayed coatings. Were the samples polished or not? The effect of adding BNNPs is clearly highlighted, but what are the reasons for such large variations in friction coefficients as they move in one direction? The authors tried to present an explanation, but it is not clearly demonstrated whether there is a correlation with the grains of the ceramic material (grain boundaries), or with the homogeneity of the distribution of the BNNPs in the surface…? To highlight these aspects (embedded BNNPs), SEM images should have included details obtained with backscattered electrons, as the secondary electrons images provide only morphology information. If it is not possible to use compo images (BS electrons), the BNNPs should have been identified by EDS.

Author Response

Reviewer #2:

At the moment, the topic is of interest and the article is written in a professional manner. However, the reviewer has identified some important issues that are not found in the article at this time.

  1. Considering the layers thickness of ~ 200 μm, in order to highlight correctly the aspects indicated in the text, the results of some analyzes performed in samples cross-section must be presented. These results can highlight the fracture mode and the depth to which they produce the effects of the force action. It is important whether the fracture is maintained in the ceramic layer or occurs in depth and even induces detachment from the metal substrate.

Response: The authors agree with the reviewer’s comment. The cross-section views of the plasma sprayed HA coatings were provided in the revised manuscript. It is clear that the coating-substrate interfaces were clear and no microcrack could be observed near the interface, implying that the adhesive strength of the coating to the substrates is high.  

  1. There are many possible logical questions for any potential reader of the article. What were the considerations used to choose the 2N and 5N forces and why higher values were not used? At what value does the exfoliation of the ceramic layer occur? What are the values of similar forces when inserting implants in cortical bone and trabecular bone?

Response: We used the applied loads of 2N, 5N and 10 N in the scratching tests, results showed that the added BNNPs are capable of mitigating the surface damage of the composite coatings at a load of 2N, 5N, respectively; the applied load increased to 10 N, the surface damage the as-sprayed HA coatings were all severe regardless of the addition BNNP nanofiller, which might ascribe to the brittle in nature of HA. Nevertheless, no exfoliation of the HA coatings was observed in this research. Generally, scratch test is usually employed to assess the bonding strength between a film and a substrate, and the film with a thickness of several microns might peel off when the applied load reaches a critical value. In this research, the thickness of the as-sprayed is around 200 μm, exfoliation of the coating with such thickness should be difficult to occur. The authors think that the reviewer’s comment of “What are the values of similar forces when inserting implants in cortical bone and trabecular bone?” is very constructive, the forces applied upon the HA coatings should be carefully analyzed in our future work based on detailed location of the implant, which should be expected to deeply understand the tribological behavior of the bioactive ceramic coatings and to satisfy the actual requirements in clinical applications.   

  1. Prior to the scratch tests, the samples were ground by 800-grit silicon carbide abrasive paper and followed by polishing with diamond particles to allow the test to be made. In real conditions of use (referred to in the introductory part of the article), this is not possible, as the implant surfaces have very complex geometries. Moreover, generally, at macro level the aim is to obtain surfaces with the highest possible roughness. Given the above, how relevant is the performed test?

Response: The authors thank the reviewer's comment. It is well known, however, that surface roughness displays significant effect on the scratch and wear bebavior of a solid material, especially on its coefficient of friction. To avoid the difference in the surface roughness of the as-sprayed HA coatings with and without the addition of BNNPs on their scratch tests, surface preparation, i.e., being ground by 800-grit silicon carbide abrasive paper and followed by polishing with diamond particles size of 1 μm, was conducted on each coating sample. Also, due to higher surface roughness associated with the plasma sprayed coating, it will usually damage the diamond tip during scratching tests.

  1. In the reviewer's opinion, performing strictly the scratch tests is not enough to achieve the proposed objectives. Performing ball bearing wear tests on the disc would have been eloquent.

Response: As we stated in the introduction of the manuscript, most wear damages of a solid material usually initiate at small length scales, and therefore the main aim of the manuscript to understand the damage mechanism of the as-sprayed HA coatings in the length scale of microns. The authors agree with the reviewer’s comment, and pin-on-disk or ball-on-disk sliding wear tests will be carried out in our future work.

  1. For a correct comparison of the results, in Figure 1 the analysis corresponding to the HA-1B sample, must be introduced.

Response: The authors thank the reviewer's comment. SEM image of surface morphology of HA-1B coating has been added in the revised manuscript.

  1. It is not clear how the measurement of the variations in coefficients of friction (CoF) was performed for the as-sprayed coatings. Were the samples polished or not? The effect of adding BNNPs is clearly highlighted, but what are the reasons for such large variations in friction coefficients as they move in one direction? The authors tried to present an explanation, but it is not clearly demonstrated whether there is a correlation with the grains of the ceramic material (grain boundaries), or with the homogeneity of the distribution of the BNNPs in the surface…? To highlight these aspects (embedded BNNPs), SEM images should have included details obtained with backscattered electrons, as the secondary electrons images provide only morphology information. If it is not possible to use compo images (BS electrons), the BNNPs should have been identified by EDS.

Response: The authors thank the reviewer's comment. The variation of CoFs of the HA coatings was measured during scratching tests using tangential friction force sensor along with micro-scratch Tester (CSM Instruments), as we described in the manuscript. The reason for such large fluctuations in residual scratch depth and friction coefficient has been discussed in the revised manuscript. Also, Raman spectroscopy was conducted on the scratch groove of the composite coatings to reveal the presence of the BNNPs, and the result was provided in the revised manuscript.

Round 2

Reviewer 2 Report

Agree.